Evaluation of the validity of the HPV viral load compared to conventional techniques for the detection of high-grade anal intraepithelial lesions in men with HIV who have sex with men

Díez-Martínez Marcos 1 2 3
Perpiñá-Galvañ Juana juana.perpina@ua.es 1 4
Ferri Joaquín 1 5
Ventero Maripaz 1 6
Portilla Joaquin 1 2 3 7
Cabañero-Martínez María José 1 4
1 Alicante Institute for Health and Biomedical Research (ISABIAL) , Alicante , Spain
2 Infectious Diseases Unit, Alicante University General Hospital , Alicante , Spain
3 Spanish AIDS Research Network, Carlos III Health Institute , Madrid , Spain
4 Nursing Department, Alicante University , Alicante , Spain
5 Surgery Department, Alicante University General Hospital , Alicante , Spain
6 Microbiology Department, Alicante University General Hospital , Alicante , Spain
7 Department of Clinical Medicine, Miguel Hernandez University , Elche , Alicante , Spain
Jose Leny
Electronic publication date: 2023 Aug 23
Publication date: 2023
Volume: 11
Electronic Location ID: e15878
Received 2023 May 15; Accepted 2023 Jul 18
Copyright: ©2023 Díez-Martínez et al.
Copyright year: 2023
Copyright holder: Díez-Martínez et al.
License: This is an open access article distributed under the terms of the Creative Commons Attribution License, which permits unrestricted use, distribution, reproduction and adaptation in any medium and for any purpose provided that it is properly attributed. For attribution, the original author(s), title, publication source (PeerJ) and either DOI or URL of the article must be cited.
License URL: https://creativecommons.org/licenses/by/4.0/

Keywords: Cytology, High-grade anal intraepithelial lesion, HIV, HPV, Viral load

Funding: Alicante Institute of Health and Biomedical Research (ISABIAL) 180213 This research was supported by the Alicante Institute of Health and Biomedical Research (ISABIAL) with the project number 180213. The funders had no role in study design, data collection and analysis, decision to publish, or preparation of the manuscript.

==============================
Background

The incidence of high-grade anal intraepithelial lesions (HSILs) has increased in recent years among men who have sex with men with human immunodeficiency virus (HIV). This work evaluated the validity of the human papilloma virus viral load (HPV-VL) versus cytological and qualitative HPV results to detect HSILs.

Methods

From May 2017 to January 2020, 93 men who have sex with men and HIV were included in an anal cancer screening program from the Infectious Diseases Unit at a tertiary-care hospital in Alicante (Spain). The gold-standard for the screening of anal HSILs is the anal biopsy using high-resolution anoscopy. The diagnostic methods compared against gold-standard were HPV-16-VL, HPV-18-VL, and HPV-16-18-VL co-testing, anal cytology, and qualitative HPV detection. The receiver operating characteristic (ROC) curve and cut-off points for HPV-VL were calculated. The sensitivity, specificity, positive predictive value (PPV), negative predictive value (NPV), and Cohen’s Kappa coefficient (κ) were also calculated.

Results

The mean patient age was 44.6 ± 9.5 years. All of them received antiretroviral treatment, 96.8% had an HIV viral load of <50 copies/mL and 17.2% had a previous diagnosis of AIDS. The diagnosis of the anal biopsies were: 19.4% (n = 18) HSIL, 29.1% (n = 27) LSIL, and 51.6% (n = 48) negative. An HPV-16-VL >6.2 copies/cell was detected in the HSIL biopsy samples (p = 0.007), showing a sensitivity of 100% and a specificity of 46.2%. HPV-18-VL and HPV16-18-VL co-testing showed a sensitivity of 75% and 76.9% and a specificity of 72.7% and 61.3%, respectively. The highest PPV was 50% obtained with the cytology and HPV-18-VL. The HPV-16-VL showed a NPV of 100%, followed by 88.9% in the HPV-18-VL and 87% in the abnormal cytology. Cohen’s Kappa coefficient were: HPV-18-VL (κ = 0.412), abnormal cytology (κ = 0.353) and HPV-16-VL (κ = 0.338).

Conclusions

HPV-VL testing improved the detection sensitivity but not the specificity for HSIL biopsies compared to anal cytology and the qualitative detection of HPV. In men who have sex with men and HIV the HPV-VL could be an useful tool for diagnosis of HSILs in anal cancer screening programs. Further studies will be needed to evaluate the clinical implications of HPV-VL in these programs.

Introduction

Anal cancer is a rare digestive tumour type whose incidence has increased in recent years (Siegel et al., 2022), especially in people with human immunodeficiency virus (HIV), and more specifically, among men who have sex with men (MSM) (Colón-López et al., 2018; Clifford et al., 2021; Koroukian et al., 2022). One of the main factors associated with the appearance of preneoplastic precursor anal cancer lesions is infection by the human papillomavirus (HPV) (Machalek et al., 2016; De Martel et al., 2017), with the risk of developing anal cancer being greater in people with HIV and HPV coinfection (Deshmukh et al., 2023). Of the precursor lesions, high-grade anal intraepithelial lesions (HSILs) have been the most widely studied (Watson et al., 2006; Machalek et al., 2012; Palefsky et al., 2022).

Anal biopsy using high-resolution anoscopy (HRA) is currently considered the gold standard for the diagnosis and management of HSIL (Richel et al., 2013; Pernot et al., 2018; Silva et al., 2018). However, various authors reject the use of HRA as a primary screening test because of poor acceptance among the population and its implementation in healthcare settings is especially difficult because it requires specialists and multiple care resources (Iribarren-Díaz et al., 2014; Lam et al., 2018; Apaydin et al., 2019; Barroso et al., 2022). For all these reasons, HRA is used as a second-line diagnostic test, with anal cytology or qualitative detection of HPV being used as the main diagnostic methods for screening for anal intraepithelial neoplasia, in line with the clinical guidelines of different national and international working groups such as the European Society for Medical Oncology, the European AIDS Clinical Society, or the AIDS Study Group (Grupo de Estudios del SIDA, 2019; EACS Society, 2020; Rao et al., 2021).

The validity of these methods for the detection of anal precursor lesions or anal cancer in people with HIV and MSM was evaluated in a meta-analysis (Clarke et al., 2022). For anal cytology with a diagnosis of atypical squamous cells of undetermined significance (ASCUS), or a more severe result, the sensitivity was 85.2% (95% CI, 77–91%) with a specificity of 52.8% (95% CI, 43–62%). Among the anal smears with a diagnosis of HSIL, the sensitivity was 24.6% (95% CI, 19–31%) and the specificity was 96% (95% CI, 93–98%). This same analysis after the qualitative detection of high-risk HPV offered a sensitivity of 96.1% (95% CI, 90–99%) and a specificity of 29.9% (95% CI, 22–39%). Finally, for the qualitative detection of HPV-16, the sensitivity was 42.4% (95% CI, 27–59%) and the specificity was 80.4% (95% CI, 74–85%).

Given the wide variability in the results published on the sensitivity and specificity of the different screening tests, new biomarkers such as HPV E6/E7 mRNA, p16, and Ki67 have been explored that could help in the screening of HSILs. The sensitivity of these tests for the detection of HSILs is also variable; for E6/E7 the sensitivity ranged from 69.6% to 71% and the specificity from 56.1% to 55.6%, while for p16 or p16/ki67, a sensitivity of 38.1% to 90% and a specificity of 50.5% to 87.9% was obtained (Wentzensen et al., 2012; Phanuphak et al., 2013a; Phanuphak et al., 2013b; Sendagorta et al., 2015; Jin et al., 2017). As we can see, there was also great variability in these results and, to date, none of these biomarkers have been included in the guidelines for routine use in anal cancer screening (Shenoy, 2022).

Determination of the HPV viral load (HPV-VL) is another biomarker directly associated with the severity of cervical intraepithelial lesions in cervical cancer screening (Chang et al., 2014; Liu et al., 2021; Zhang et al., 2022), with the HPV-16-VL also correlating with oropharyngeal cancer (Biesaga et al., 2018), among others. In the population with HIV, HPV-VL also appears to predict anal intraepithelial neoplasia (Poizot-Martin et al., 2009), HSIL lesions in people with an HPV-16-VL ≥ 65 copies/cell (Agsalda-Garcia et al., 2018), and tumour control in patients with a low tumour viral load (Rödel et al., 2015). There is also evidence for an association between a high HPV-16-VL and the absence of previous chemotherapy or radiotherapy treatments in patients with anal cancer (Małusecka et al., 2020). On the contrary, some studies associate a high HPV viral load with better overall survival in people with anal cancer (Guerendiain et al., 2022) and so the current evidence regarding the relationship between HPV-VL and anal cancer is inconclusive. On the other hand, although the prevalence of HPV-18 relative to HPV-16 is lower in the diagnosis of anal lesions, it is also detected in anal cancer (Alemany et al., 2015). However, because of this lower prevalence, no previous studies have yet analysed HPV-18-VL in cases of HSIL.

To the best of our knowledge, HPV-VL has not been studied in the population of MSM with HIV in Spain. Therefore, considering the good predictive results of the majority of published international studies, we decided to study the ability of the HPV-VL to detect high-grade anal lesions in a population of MSM with an HIV infection who continued clinical monitoring at our clinic. The objective of this study was to evaluate the validity of using the HPV-VL to detect HSILs in men with HIV who have sex with men compared to the efficacy of qualitative and cytological HPV test results.

Material & Methods

Design

This was a cross-sectional study of a diagnostic test evaluation.

Population

Eligible candidates for anal cancer screening were included from the Infectious Diseases Unit at a tertiary-care hospital in Alicante (Spain). Men who attended an anal cancer screening test between May 2017 and January 2020 were consecutively selected to participate in this study. All of them met the following inclusion criteria: males aged 18 years or older, a confirmed HIV infection, an HIV transmission mechanism of MSM sexual relations, and participating in the anal cancer screening program. The only exclusion criterion was having been previously vaccinated against HPV.

Study variables and data collection

An ad hoc structured data collection questionnaire was prepared for data collection. Sociodemographic variables (sex and age), HIV-related clinical variables (infection transmission mechanism, previous diagnosis of acquired human immunodeficiency syndrome [AIDS], antiretroviral treatment (ART), undetectable HIV viral load [HIV-RNA <50 copies/mL], CD4+ nadir, current CD4+ cell value/µL, CD4/CD8 ratio and presence of genital warts), toxic habits (smoking, alcohol use, or recreational drug use), and variables related to sexual habits (age in years at the time of the first complete sexual relationship, number of sexual partners in the year prior, lifetime number of sexual partners, condom use, and history of sexually transmitted diseases) were collected.

Anal cytology

Anal cytology was performed on all the participants using the specific cytobrush material which was later transferred to ThinPrep sample preservation jars®. This technique involves introducing a brush into the anal canal using enveloping movements from the perianal skin up to four cm above the anal margin. Subsequently, this brush was introduced into a liquid medium for a few seconds (ThinPrep™, PreservCyt Pap Test™; Hologic Corp, Marlborough, MA, USA) for preservation and analysis. Analysis of the anal cytology results was first performed by a single pathologist specialising in the diagnosis of anal dysplasia. The Bethesda classification was used (Solomon et al., 2002) with the results being: negative, ASCUS, low-grade squamous intraepithelial lesion (LSIL), or high-grade squamous intraepithelial lesion (HSIL).

The anal cytology was subsequently sent to the clinical microbiology service to determine the viral load and qualitative detection of HPV. The viral load of HPV-16 and HPV-18 was determined and the viral load co-testing of HPV-16-18 was performed taking the highest quantified value of HPV-16 or HPV-18 as the reference. Detection of human genomic DNA in the HPV viral load was performed using TaqMan™ Control Genomic DNA (Thermo Fisher Scientific, Waltham, MA, USA). The results were normalised and expressed as viral copies/human cell (copies/cell). The qualitative detection of HPV was carried out with the Cobas® 4800 HPV-Test. The results obtained were for HPV-16 and HPV-18, as well as a combined method for 12 high-risk human papillomavirus genotypes (HR-HPV), including 31, 33, 35, 39, 45, 51, 52, 56, 58, 59, 66, and 68.

High resolution anoscopy

The HRA was performed on all the participants by the same team which comprised a proctologist and a nurse, both trained in performing this technique. The anoscopy was carried out in an operating room set up for this purpose. First, a digital rectal examination and visual examination were performed to detect macroscopic changes. Second, a transparent plastic anoscope (THD® N-Ano) was introduced into the anal canal for visualisation purposes. Using a colposcope with a light source and high magnification binocular vision, the anal tissue was then visualised. Afterwards, a swab wrapped in gauze and impregnated with 3% acetic acid (Lugol’s stain) was introduced for 2 min and was withdrawn; the anoscope was then reintroduced to visualise the anal tissue. After staining, areas likely to be preneoplastic—acetowhite positive—were sought. When necessary, a further 5% Lugol staining was performed with a swab to improve visualisation of the possible lesions. Third, an anal biopsy was taken from areas suspected of a lesion using Baby Tischler forceps and was then sent to the pathology department for analysis. When a patient had multiple areas biopsied on the same day, the area with the highest degree of dysplasia was used for analysis. The histological classification of the anal biopsy for its diagnosis was based on the Lower Anogenital Squamous Terminology Project (Darragh et al., 2012), with the following possible results: negative biopsy, LSIL, and HSIL.

Procedure

All the participants were informed of the reason for the screening and signed an informed consent form for their participation in the study and for data collection. Participants were informed during HIV medical follow-up visits of the simultaneous performance of: anal cytology, qualitative HPV detection and high-resolution anuscopy with anal biopsy. These results were analysed by the Pathological Anatomy Service. The Clinical Microbiology Service subsequently analysed the viral load and carried out the qualitative detection of HPV.

Statistical analysis

A descriptive analysis of the continuous variables was performed using the mean and standard deviation (SD) and of the categorical variables using the absolute frequencies and percentages. Anal biopsy using HRA was used as the gold standard method of screening for HSILs. In the statistical analysis of the association and comparison of the different diagnostic tests, the histological result of HSIL versus non-HSIL was used. The latter included negative anal biopsy, LSIL, and anal warts. Thus, the continuous variables were analysed using Student t-tests or Mann–Whitney U tests, depending on the normality of the variables (calculated using the Kolmogorov–Smirnov test). Categorical variables were analysed using chi-squared tests and Fisher tests for association studies. In all the analyses, the value of p < 0.05 was considered statistically significant.

Finally, the receiver operating characteristic (ROC) curves, together with the area under the curves (AUCs), were calculated to detect the cut-off point of the viral load of HPV-16, HPV-18, and the HPV-16-18 co-test to differentiate between the diagnosis of HSILs and non-HSILs by HRA biopsy. Sensitivity, specificity, positive predictive value (PPV), negative predictive value (NPV), and Cohen’s Kappa coefficient (κ) were calculated to compare the usefulness of the different screening methods: HPV-16, HPV-18, HPV-16-18 co-testing, anal cytology, and the qualitative determination of HPV. Taking into account the high prevalence of anal HSILs data found in other studies (Phanuphak et al., 2013a; Phanuphak et al., 2013b; Burgos et al., 2017; Agsalda-Garcia et al., 2018), it seemed appropriate to calculate the values of PPV and NPV with a prevalence of 20%. All the data analyses were performed with using SPSS software (version 21; IBM Corp., Armonk, NY, USA).

Ethical considerations

The study was approved at the Hospital General Universitario Alicante clinical ethics committee (project reference code UGP-18-249). All the participants were informed about the study and authorisation was requested by written informed consent before their participation. The confidentiality and anonymity of the data was ensured in accordance with current legislation on the protection of personal data and guaranteed rights (LOPD 03/2018 of December 5) and EU Regulation 2016/679 of the European Parliament.

Results

From May 2017 to January 2020, a total of 103 patients were seen at a tertiary-care hospital anal cancer screening clinic: 93 men and 10 women. A total of 93 participants who were MSM with an HIV infection were included in this work. Sociodemographic variables, clinical variables related to HIV, toxic habits, and sexual habits were collected of all the participants (Table 1). The mean patient age was 44.6 ± 9.5 years, they all received antiretroviral treatment, and 17.2% had a previous diagnosis of AIDS. The mean CD4+ count was 815 ± 357 cells/µL and 96.8% had an HIV viral load of <50 copies/mL; 40.9% of the participants smoked and 29% used recreational drugs on a regular basis. The median age of the first sexual relationship was 16 years, 51.6% did not use a condom in their relationships, and 67.7% had had at least one sexually transmitted disease.

Table 1 Sociodemographic variables related to HIV, toxic habits, and sexual habits (N = 93).

	n (%)	
Sociodemographic variables		
Age, years, mean (SD)	44.6 ± 9.5	
HIV-related clinical variables	
MSM	93 (100)	
Previous AIDS	16 (17.2)	
ART, n (%)	93 (100)	
HIV-VL <50 copies/mL, n (%)	90 (96.8)	
Nadir CD4+, mean (SD)	389 ± 264	
CD4 cell/ μL, mean (SD)	815 ± 357	
CD4/CD8 ratio, mean (SD)	0.87 ± 0.4	
Presence of condyloma, n (%)	47 (50.5)	
Toxic habit variables	
Tobacco use, n (%)	38 (40.9)	
Alcohol consumption, n (%)	55 (59.1)	
Recreational drug use, n (%)	27 (29)	
Variables related to sexual habits	
Age at the time of first relationship, median (IQR)	16 (14–18)	
Sexual contacts over the last year, median (IQR)	5 (1–20)	
Lifetime sexual contacts, median (IQR)	100 (30–200)	
Non condom use, n (%)	48 (51.6)	
Previous STI, n (%)	63 (67.7)	
Notes.

AIDS Acquired immunodeficiency syndrome

ART antiretroviral therapy

HIV human immunodeficiency virus

IQR interquartile range

MSM men who have sex with men

SD standard deviation

STI sexually transmitted infection

A descriptive analysis of the different diagnostic tests studied for anal cancer screening was performed using HSIL and non-HSIL anal HRA biopsy for comparison (Table 2). In the anal biopsy, 19.4% (n = 18) presented a HSIL, 29.1% (n = 27) a LSIL, and 51.6% (n = 48) a negative result. The quantification of the viral load in the HPV-16-18 co-test was higher for the total number of participants compared to HPV-16 and HPV-18 (41 copies/cells vs. 28.9 copies/cells vs. 2.16 copies/cells). In the anal cytology, 17.2% (n = 16) of the participants had a cytological alteration (LSIL 10.8%, ASCUS 2.2%, and HSIL 4.3%) while 81.7% (n = 76) obtained a negative result. There was one patient whose cytological result could not be obtained because insufficient material had been obtained. The highest prevalence of HPV detected was HR-HPV accounting for 83.7% (n = 77), followed by HPV-16 at 40.2% (n = 37), and finally HPV-18 with 17.4% (n = 16). In the comparisons made between participants with HSIL versus non-HSIL anal biopsy results, there were significant differences in HPV-16-VL (198 copies/cells vs. 16.3 copies/cells; p = 0.036) and the HPV-16-18 co-test viral load (205 copies/cells vs. 25 copies/cells; p = 0.049). In the rest of the variables studied, significant differences were found in abnormal anal cytology (p = 0.001), qualitative HPV-16 (p = 0.044), and qualitative HR-HPV (p = 0.037).

Table 2 Descriptive and comparative analysis for anal cancer screening between participants with HSIL and non-HSIL anal biopsy results (N = 93).

	Total	HSIL	non-HSIL	p-value	
HPV-16-VL* (copies/cells), p50 (IQR)	28.9 (2.8–514)	198 (20.7–1,342)	16.3 (0.9–228)	0.036	
HPV-18-VL (copies/cells), p50 (IQR)	2.16 (0.04–562)	880 (20.7–1,728)	0.5 (0.03–157)	0.177	
HPV-16-18-VL (copies/cells), p50 (IQR)	41 (2–822)	205 (32–1,712)	25 (1–336)	0.049	
Abnormal cytology**, n (%)	16 (17.2)	8 (44.4)	8 (10.7)	0.001	
ASCUS	2 (2.2)	0 (0)	2 (2.7)		
LSIL	10 (10.8)	4 (22.2)	6 (8)		
HSIL	4 (4.3)	4 (22.2)	0 (0)		
Negative	76 (81.7)	10 (55.6)	66 (88)		
Insufficient material	1 (1.1)	0 (0)	1 (1.3)		
Qualitative HPV-16, n (%)	37 (40.2)	11 (61.1)	26 (35.1)	0.044	
Qualitative HPV-18, n (%)	16 (17.4)	4 (22.2)	12 (16.2)	0.508	
Qualitative HR-HPV***, n (%)	77 (83.7)	18 (100)	59 (79.7)	0.037	
Notes.

* HPV-VL, viral load of the human papilloma virus; IQR, interquartile range.

** Abnormal cytology, atypical squamous cells of undetermined significance (ASCUS); low-grade squamous intraepithelial lesion (LSIL); high-grade squamous intraepithelial lesion (HSIL).

*** HR-HPV, high-risk human papilloma virus genotypes 31, 33, 35, 39, 45, 51, 52, 56, 58, 59, 66, and 68.

The results regarding the validity of HPV-VL, anal cytology, and qualitative HPV detection for the diagnosis of anal HSIL in participating HIV positive MSM were compared (Table 3). The best cut-off points were calculated using the ROC curve in the viral load of HPV-16, HPV-18, and in the HPV-16-18 co-test for the detection, using the HRA anal biopsy of the HSIL taken in the study as the gold standard. When detecting HSILs (Fig. 1) the following values were obtained: a HPV-16-VL of 6.2 copies/cell (AUC = 0.720); 59.8 copies/cell (AUC = 0.750) for HPV-18-VL; and a value of 41.7 copies/cell for HPV-16-18-VL co-testing (AUC = 0.690). The sensitivity, specificity, PPV, and NPV of the tests for determining the presence of HSILs were calculated using these aforementioned results. The sensitivity and specificity of abnormal anal cytology, which is normally used for diagnosing HSIL, was 44.4% and 89.3%, respectively. HPV-16-VL showed a sensitivity of 100% with a specificity of 46.2%; however, the latter improved slightly in the HPV-16-18-VL co-testing, showing a sensitivity of 76.9% and a specificity of 61.3%. HPV-18-VL showed the results with the lowest quantified difference between sensitivity and specificity, at 75% and 72.7%, respectively. The qualitative detection of HPV-16 showed a sensitivity of 61.1% and a specificity of 64.9%. The HPV-18-VL (κ = 0.412) showed the highest Cohen’s Kappa coefficient followed by abnormal cytology (κ = 0.353) and HPV-16-VL (κ = 0.338). However, the poorest results were in the qualitative HR-HPV (κ = 0.037) and the qualitative HPV-18 (κ = 0.063).

Table 3 The validity of HPV viral load, anal cytology, and qualitative HPV detection for the diagnosis of anal HSILs in HIV positive MSM (N = 93).

Screening tests evaluated	Anal histology for HSIL	
	Sensitivity (%)	Specificity (%)	PPV (%)	NPV (%)	p-value	Kappa	
Abnormal cytology*	44.4	89.3	50	87	0.001	0.353	
Qualitative HR-HPV**	77.8	29.7	21.2	84.6	0.526	0.037	
Qualitative HPV-16	61.1	64.9	29.7	87.3	0.044	0.186	
Qualitative HPV-18	22.2	83.8	25	81.6	0.508	0.063	
HPV-16-VL	100	46.2	44	100	0.007	0.338	
HPV-18-VL	75	72.7	50	88.9	0.235	0.412	
HPV-16-18-VL co-testing	76.9	61.3	45.5	86.4	0.021	0.318	
Notes.

HPV-VL viral load of the human papilloma virus

NPV negative predictive value

PPV positive predictive value

* Abnormal cytology: atypical squamous cells of undetermined significance (ASCUS); low-grade squamous intraepithelial lesion (LSIL); high-grade squamous intraepithelial lesion (HSIL); relative risk (RR).

** HR-HPV: high-risk human papilloma virus genotypes 31, 33, 35, 39, 45, 51, 52, 56, 58, 59, 66, and 68.

Figure 1 Receiver operating characteristic (ROC) curve and its corresponding area under the curve (AUC) used for the diagnosis of high-grade intraepithelial lesions (HSILs) using the viral load of HPV-16, HPV-18, and HPV-16-18 co-testing.

Discussion

The aim of this study was to evaluate the validity of HPV-VL determination, compared to cytological and qualitative HPV results, for the diagnosis of HSILs using the histological results of anal biopsies as the gold standard. HPV-16-VL showed the highest sensitivity for the diagnosis of HSILs in anal cancer screening, although with a low specificity. HPV-18-VL showed the most balanced results between sensitivity and specificity with the drawback that only a few participants were studied. However, HPV-16-18-VL co-testing produced the most similar sensitivity and specificity results compared to conventional methods of anal cytology and qualitative HPV detection.

As already observed in previous studies in HIV-infected men (Palefsky et al., 2005; Gaisa et al., 2014; Machalek et al., 2016), the anal biopsy produced a pathological result in almost half the participants (48.5%). However, only slightly more than 17% of anal Papanicolaou smears produced pathological results. This discrepancy between abnormal cytology and histology results has been previously described and may be because of inter-observer variability in the interpretation of both anal samples (Nathan et al., 2010; Weis et al., 2011; Iribarren Díaz et al., 2017). Despite the lower frequency of pathological results from anal cytologies compared to anal biopsies, there was a statistically significant difference between them for the diagnosis of HSILs with abnormal cytology compared to non-HSILs.

In contrast, in our sample there was a high prevalence of qualitative HPV. The prevalence of HPV-16 was around 40%, similar to that found in other studies (Palefsky et al., 1998; Poizot-Martin et al., 2009; Van Aar et al., 2013; Hernandez et al., 2016; Beliakov et al., 2021), which was almost twice as high in HSILs as in non-HSILs. Even so, the involvement of other HPVs in HSILs cannot be ruled out given that in our study, HR-HPV was qualitatively detected in 100% of HSILs compared to 79.7% of non-HSILs.

In turn, in our work, abnormal anal cytology also showed low sensitivity with respect to the quantification of HPV-16 or HPV-16-18 co-testing (44.4% vs. 100% vs. 76.9%, respectively) but high specificity (89.3% vs. 46.2% vs. 61.3%, respectively). A meta-analysis reported a detection sensitivity of 85.2% and a specificity of 52.8% for anal cytology of HSILs (Clarke et al., 2022). The low sensitivity for anal cytology in our study could be because of intra-observer variability in the interpretation of the results (Lytwyn et al., 2005). Every case diagnosed with a HSIL biopsy had an HPV-16-VL >6.2 copies/cell, representing figures lower than those found in other studies (Agsalda-Garcia et al., 2018). However, the different ways of quantifying the viral load used in other studies, such as the number of HPV-DNA copies/ng of the total DNA in the sample (Pierangeli et al., 2008) or the viral load of the β-globin gene (Rödel et al., 2015) makes it difficult to compare the results. In general, the results, especially from our study, showed that HPV-16 is strongly implicated in high-grade anal dysplasias, with lower quantification for HPV-VL compared to other viral genotypes.

Nonetheless, the sensitivity of the HPV-16-18-VL co-test was lower than that of the HPV-16-VL (76.9% vs. 100%), although the specificity was still improved (61.3% vs. 46.2%). It is possible that including HPV-18-VL with HPV-16-VL to obtain a combined result will improve the diagnostic validity of viral load in the diagnosis of HSILs. In addition, HPV-18-VL produced results with a lower discrepancy between sensitivity and specificity, at 75% and 72.7%, respectively. However, these data must be interpreted with caution because very few cases with HPV-18-VL and HSILs have been studied, and so these results are difficult to extrapolate. Notwithstanding, other studies have suggested that HPV-18 could also be a useful biomarker for the detection of anal lesions (Hidalgo-Tenorio et al., 2021).

Of note, this current work had several limitations. Firstly, there could have been a participant selection bias because all the individuals who attended the anal cancer screening consultation were included consecutively. The fact that these patients were attending this consultation was conditioned by the fact that they had a greater interest in undergoing screening for possible anal symptoms suggestive of disease. Secondly, the low number of participants with qualitative HPV-18 could have influenced the interpretation of the results because these data cannot be extrapolated to other contexts and so more patients must be studied in future work.

However, despite the limitations of this study, this work is the first of its type to be carried out at a national level in Spain to evaluate and compare the validity of HPV-VL to detect HSILs to that of conventional techniques in anal cancer screening in a population with HIV. While other studies have screened participants with a prior pathological result in anal cytology (Iribarren-Díaz et al., 2014; Agsalda-Garcia et al., 2018; Pernot et al., 2018), with possible infection by HPV or other pathogens between the time of the cytology and anal biopsy, in our study no prior cytology was conducted, and all the procedures were performed simultaneously. We attempted to control for interobserver bias in the diagnosis of anal biopsy by having the same pathologist examine all the anal biopsies sent to the Pathology Service. Lastly, we included HPV-18-VL, which had not been previously studied elsewhere.

Given the wide variability in the results obtained for the predictive capacity of conventional methods such as anal cytology and the qualitative detection of HPV, studying new methods such as HPV-VL quantification can help in the detection of HSILs. This technique is a minimally invasive, well-tolerated test that also eliminates inter-observer variability in the interpretation of anal cytology. Given the overall high sensitivity of viral load quantification reported in this current work, perhaps in the future this technique could be considered a good biomarker for anal cancer screening.

Conclusions

In conclusion, in our work, quantification of HPV with cut-off values for HPV-16, HPV-18, and HPV-16-18 co-testing of 6.2, 59.8, and 41.7 copies/cell, respectively, improved the sensitivity for the detection of HSILs compared to anal cytology or the qualitative detection of HPV, although it did not improve the specificity. In addition, the HPV-16-VL and HPV-16-18-VL co-testing results were statistically significant for the diagnosis of HSILs versus non-HSILs. All of this reinforces the idea that HPV is implicated in anal cancer and can be used for screening and detection of high-grade lesions. Nonetheless, further studies with more participants will be needed to evaluate the clinical implications that HPV-VL quantification may have in anal cancer screening. Indeed, it would be interesting to conduct longitudinal studies that evaluate HPV-VL at different screening time points in the same patients in order to study the factors that interfere with the progression of the HSILs or the clearance of HPV itself.

Supplemental Information

Supplemental Information 1 HPV viral load data

Click here for additional data file.

Many thanks to José Sánchez for advice on the study design and data analysis. In addition, we would like to thank Juan Carlos Rodriguez and Adelina Gimeno for allowing us to develop new molecular microbiology tests in the microbiology department in the Hospital General Universitario de Alicante.

Additional Information and Declarations

Competing Interests

Author Contributions

Human Ethics

Data Availability

The authors declare there are no competing interests.

Marcos Díez-Martínez conceived and designed the experiments, performed the experiments, analyzed the data, prepared figures and/or tables, and approved the final draft.

Juana Perpiñá-Galvañ conceived and designed the experiments, prepared figures and/or tables, authored or reviewed drafts of the article, and approved the final draft.

Joaquín Ferri performed the experiments, authored or reviewed drafts of the article, and approved the final draft.

Maripaz Ventero performed the experiments, analyzed the data, prepared figures and/or tables, and approved the final draft.

Joaquin Portilla conceived and designed the experiments, analyzed the data, authored or reviewed drafts of the article, and approved the final draft.

María José Cabañero-Martínez conceived and designed the experiments, prepared figures and/or tables, authored or reviewed drafts of the article, and approved the final draft.

The following information was supplied relating to ethical approvals (i.e., approving body and any reference numbers):

The study was approved by a clinical ethics committee Hospital General Universitario Alicante (project number ugp-18-249).

The following information was supplied regarding data availability:

All data analysed about HPV viral load are available in the Supplemental Files.

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
