# Peer review of "Evaluation of the validity of the HPV viral load compared to conventional techniques for the detection of high-grade anal intraepithelial lesions in men with HIV who have sex with men"

_PeerJ, doi:10.7717/peerj.15878_

## Round 0.1 · original submission · Minor Revisions

Two reviewers raised some minor concerns. Please address all comments comprehensively and resubmit.

·

Basic reporting

No comment.

Experimental design

Abstract
1. The study setting is not provided. Where was the study conducted?
2. In the methods section, clearly state the screening approaches being evaluated and the gold-standard they are being evaluated against.
3. In the opening sentence in the methods (line 25), avoid starting a stance with figures (93).
4. Results: too brief; it would be valuable to add some descriptive information about the 93 men. Also, the results for the predictive values and kappa are missing.
5. Conclusion: even before other studies are conducted, what would you recommend from your study? To adopt HPV-VL in this high-risk population or not?

Main body
Methods
1. It appears the study sample size was based on a defied time period as opposed to simulations for minimum sample size required. What was the basis for selecting the four-year period? When was the study itself conducted, in relation to the four-year period (2017-2020)? If the men had undergone screening, why they recalled to undergo further evaluation in the study? If yes, what screening test had been used before? Or alternatively, was the study running prospectively for the four years, so that all the tests mentioned in the study were conducted during the routine screening of the participants? Understanding this context is important to be able to put the results in perspective.
2. Statistical analysis: to my understanding, calculating PPV and NPV requires prevalence of the study variable in the population. Which prevalence value did you adopt?

Results
I have not seen the results for the Cohen’s Kappa coefficient mentioned in the results, although it is in table 3.

Validity of the findings

No comment.

Additional comments

Overall comments
This is a very well written paper on an important topic. Although the sample size was small, it provides important information for possible bio-markers for population-level anal cancer screening.

Conclusion
The authors rightfully state the promise of HPV-VL as a screening modality at population-level for anal cancer. They also recommend longitudinal studies to evaluate the bio-marker at several screening points. Since this study is based on a structured screening program in Spain, I would propose that this study is transformed into the longitudinal study, alongside others that may be implemented in future.

Reviewer 2 ·

Basic reporting

No comments

Experimental design

No comments

Validity of the findings

No comments

Additional comments

The manuscript titled “Evaluation of the validity of the HPV viral load compared to conventional techniques for the detection of high-grade anal intraepithelial lesions in men with HIV who have sex with men” authored by Marcos Diez-Martinez, Juana Perpiña Galvañ, Joaquin Ferri, Maripaz Ventero, Joaquin Portilla and Maria Jose Cabañero-Martinez submitted to PeerJ attempts to evaluate the validity of using HPV viral load to detect high grade anal lesions in HIV positive men who have sex with men. Though limited to a sample size of just 93 men, consecutively recruited, in my opinion, the authors have organized the study well and have justified their findings effectively. It is well appreciated that the authors have rightly addressed the limitations of the study. Apart from the concern that the clarity of figure1 needs to be improved as the axis labels in the current version cannot be read, I have no other suggestions and would recommend that this manuscript be “accepted” for publication in PeerJ.

---

## Round 0.2 · accepted · Accept

The reviewer comments have been addressed satisfactorily. I recommend publication.